# Harmonic Phasor Estimation Method Considering Dense Interharmonic Interference

**DOI:** 10.3390/e25020236

**Published:** 2023-01-27

**Authors:** Xianyong Xiao, Runze Zhou, Xiaoyang Ma, Rui Xu

**Affiliations:** The College of Electrical Engineering, Sichuan University, Chengdu 610065, China

**Keywords:** harmonic analysis, interharmonics, harmonic phasor estimation, dense frequency signal

## Abstract

Due to the limitation of frequency resolution and the spectrum leakage caused by signal windowing, the spectrums of harmonic and interharmonic components with close frequencies overlap each other. When the dense interharmonic (DI) components are close to the harmonic spectrum peaks, the harmonic phasor estimation accuracy is seriously reduced. To address this problem, a harmonic phasor estimation method considering DI interference is proposed in this paper. Firstly, based on the spectral characteristics of the dense frequency signal, the phase and amplitude characteristics are used to determine whether DI interference exists in the signal. Secondly, an autoregressive model is established by using the autocorrelation of the signal. Data extrapolation is performed on the basis of the sampling sequence to improve the frequency resolution and eliminate the interharmonic interference. Finally, the estimated values of harmonic phasor, frequency and rate of change of frequency are obtained. The simulation and some experimental results demonstrate that the proposed method can accurately estimate the parameters of harmonic phasors when DIs exist in the signal, and has a certain anti-noise capability and dynamic performance.

## 1. Introduction

With a large amount of renewable energy and nonlinear loads connected to the grid, harmonics and interharmonics are becoming increasingly abundant, posing a great threat to the safe and stable operation of the grid [1,2]. On the one hand, some large industrial equipment can generate dense interharmonics (DIs) with close frequencies [3]. At the same time, many harmonic components are considered as interharmonics due to the asynchronous sampling [4]. When these signals are mixed together, serious spectral interference is generated, leading to distortion of the measurement of harmonic parameters. Therefore, it is significant to study the estimation method for harmonic phasor under DI interference.

At present, the estimation methods of harmonic phasor can be divided into two categories: non-signal-transformation-based and signal-transformation-based methods. The non-signal-transformation-based methods include model-based and mode-decomposition-based methods. Among them, the model-based methods, such as the Prony algorithm [5], least squares method and rotational invariance technique (ESPRIT) [6], are employed. On the one hand, these methods have a deficiency of adaptive capability and cannot follow the dynamic signal to change the model coefficients in time. On the other hand, these methods will cause large errors when the signal to noise ratio (SNR) is low. The mode-decomposition-based methods include Hilbert transform (HT) and variational mode decomposition (VMD) [7]. However, these methods require a pre-set number of modes to be decomposed. If the number is not set properly, the decomposition results will be biased.

The signal-transformation-based methods can be further divided into static signal estimation and dynamic signal estimation methods. The static signal estimation methods include discrete Fourier transform (DFT), S-transform, wavelet transform, etc. Among them, DFT is simple to implement, has better stability and suitability, and has gained wide application [8]. However, it cannot meet the requirements for dynamic signal detection. To achieve dynamic signal estimation, some improved DFT methods have been proposed, such as compressive sensing-DFT (CSDFT) [9], interpolated-DFT (IpDFT) [10], improved interpolated dynamic-DFT (IIpD2FT) [11], etc. Among them, CSDFT can improve the space of the frequency grid without greatly prolonging the total observation time. IpDFT can achieve accurate harmonic phasor estimation under large fundamental frequency deviation. IIpD2FT can adequately suppress the interference of the second harmonic in various cases. These methods can satisfy the accuracy requirements of IEEE standards for P-class and M-class phasor measurement units (PMUs) under various dynamic conditions. However, the estimation errors of these methods exceed the measurement requirements for M-class PMUs under the condition where out-of-band interference exists [12,13].

The estimation methods for dynamic signals have been studied more frequently. Reference [14] extended the concept of phasor from static to dynamic and proposed an algorithm to approximate the dynamic phasor by second-order Taylor polynomials. However, the length of the observation window used in this method is short and vulnerable to the effects of noise and interharmonics. Reference [15] proposed the Taylor–Fourier transform (TFT) and extended the coefficients of the TFT to include not only the instantaneous values of harmonic fluctuations, but also the first-order derivatives in the Taylor approximation. However, it still could not get rid of the influence of wide-band noise. To suppress the interference of interharmonics and noise, [16,17,18] proposed a Taylor–Fourier multi-frequency (TFM) model based on compressive sensing (CS), which has a high accuracy under severe operating conditions, such as interharmonics, harmonics and noise. However, it has a high complexity when the voltage/current signal contains a large number of frequency components, so it cannot meet the requirements of the IEEE standard for the algorithm response time. To reduce the computational burden of CS-TFM, [19] reduces the complexity by simplifying the process of finding the frequency components, which can achieve a sufficiently short processing time in the analysis of multi-frequency phasors. However, this method requires a predetermined order of the TFM model, which can lead to large estimation errors if the order is not set properly. Other TFM-based methods include the improved iterative TFM (I2TFM) [20] and the CS-TFM with Kalman filter (KF) to track the frequencies of harmonic and interharmonic components [21]. These methods have better detection results for components with small amplitudes, but these methods only consider the lower harmonics. References [14,15,16,17,18,19,20,21] are based on the Taylor formula to model the phasor. References [22,23] proposed a harmonic phasor estimator based on the sinc interpolation function (SIFE), which utilizes the sinc function polynomial instead of the Taylor polynomial in the TFT and TFM. Under the conditions of frequency deviation, harmonic oscillation and frequency ramp, SIFE is more accurate in the estimation of harmonic phasor, frequency and rate of change of frequency (ROCOF). However, this method requires extensive simulations to select the model parameters, and zero-error results cannot be obtained at nominal frequencies. To improve this, a harmonic phasor estimator based on the frequency-domain sampling theorem was proposed in [24], which models harmonic phasors using an imaginary exponential function. This method overcomes the above-mentioned drawbacks and provides high accuracy under frequency deviation and modulation.

All of the above methods can be used for harmonic phasor estimation, but accuracy may be lost when DI components exist in the voltage/current signal. In fact, complex power signals often contain interharmonics with very close frequencies, which has an adverse effect on harmonic phasor estimation. Reference [25] proposed a spectrum separation and parameter measurement algorithm for multiple interharmonics, where each observed spectral peak is separated to obtain several frequency components. However, this method is sensitive to noise and its estimation order limits the number of frequency components obtained. Reference [26] introduced the Chirp-Z transform (CZT) method, which serves to refine the DFT spectrum. However, when spectral interference occurs, the CZT can only amplify the distorted spectrum and cannot eliminate the interference effect. Some modern optimization algorithms, such as genetic algorithms, particle swarm algorithms [27], and neural networks [28], although they have a high accuracy, all require some prior information and their applications are restricted.

To address the above problems, this paper proposes a harmonic phasor estimation method considering DI interference. Firstly, the estimation model for harmonic phasor is established. Based on the characteristics of the spectral lines with spectral interference, we use the phase and amplitude characteristics of the spectral lines to determine whether there are DIs in the signal. Secondly, for the signal containing DI interference, an autoregressive model is established using the sampled values based on the autocorrelation of harmonic data. Moreover, data extrapolation is performed to improve the frequency resolution, so that all interharmonic components can be estimated and eliminated. Finally, the estimates of harmonic phasor, frequency and ROCOF are calculated. Simulation and experimental results demonstrate that under the conditions of asynchronous sampling and DI interference, the proposed method can accurately estimate harmonic phasors. In addition, it has a certain dynamic performance and anti-noise capability, and presents superiority in comparison with other methods. The proposed method belongs to the dynamic signal estimation methods. Since the Taylor–Fourier transform is used, the proposed method can also be classified as signal-transformation-based methods.

This paper is structured as follows. Section 2 presents the estimation model for harmonic phasors and the interference caused by DIs. Section 3 describes the proposed method in detail. Section 4 shows the simulation and experimental results. Finally, Section 5 concludes the whole paper.

## 2. Harmonic Phasor Estimation Model and Existing Problems

### 2.1. Harmonic Phasor Estimation Model

In general, the actual signal containing harmonics can be represented in the time domain as
(1)$$s(t)=\sum_{h=1}^{H}a_{h}(t)\cos(2\pi f_{h}t+\varphi_{h}(t))$$
where *a_h_*(*t*) and *φ_h_*(*t*) are the amplitude and phase of the *h*-th harmonic, respectively. *f_h_* is the harmonic frequency. *H* is the maximum number of harmonics. According to the IEEE standard [29], the harmonic phasor *P_h_*(*t*) corresponding to *f_h_* is expressed as follows:(2)$$P_{h}(t)=\frac{a_{h}(t)}{\sqrt{2}}e^{j\varphi_{h}(t)}$$

The Taylor–Fourier (TF) model can describe the changes of amplitude and phase in the observation time. Therefore, a Taylor expansion model of order *K_h_* is used to represent *P_h_*(*t*). By substituting the phasor *P_h_*(*t*) after the Taylor expansion into the expression of *s*(*t*), we can obtain
(3)$$s(t)=\sum_{h=1}^{H}\sum_{k=0}^{K_{h}}\frac{t^{k}}{k!}\left[\frac{p_{h}^{\left(k\right)}}{\sqrt{2}}e^{j2\pi f_{h}t}+\frac{p_{h}^{*(k)}}{\sqrt{2}}e^{-j2\pi f_{h}t}\right]$$
where *p_h_*^(*k*)^ is the TF coefficient, which is the *k*-th order derivative of *P_h_*(*t*) at time *t* = 0. * is the conjugate operator. For different *h*-th harmonics, the order of Taylor expansion may be different. Equation (3) establishes the TFM model, which provides the basis for various analyses, including the estimation of harmonic phasors.

### 2.2. Interference of DIs on Harmonic Phasor Estimation

DIs can be understood as interharmonics with close frequencies. Due to the signal windowing in the time domain, DFT produces spectral leakage, leading to the appearance of the main and side lobes of each frequency component. When two adjacent spectral peaks of interharmonics are close to each other, the main and side lobes of spectral peak A are overlapped onto the main and side lobes of spectral peak B. This will cause serious spectral interference problems and thus lead to deviations in the positions of the spectral peaks of the dense components in the spectrum. Especially when the interharmonic spectral peaks are close to the harmonic spectral peaks, the estimation accuracy of the harmonic phasor will be seriously affected.

Figure 1 shows the spectrum diagram of the signal *x*(*t*) = 0.5cos(2π × 95.5*t*) + cos(2π × 100*t*) + 0.5cos(2π × 102.8*t*). The sampling frequency is 5 kHz and the number of sampling points is 1024. From Figure 1, two DIs exist near the 2nd harmonic, causing serious spectral interferences. Only two spectral peaks can be observed, and their corresponding frequencies and amplitudes deviate significantly from the true values. In order for the calculated spectrum to reveal the adjacent spectral peaks, it is necessary that the frequency difference between the two adjacent components is greater than or equal to the frequency resolution. Although this could be achieved by increasing the number of sampling points, that is, increasing the observation interval, this would reduce the dynamic tracking capability for the signal. The proposed method can improve the frequency resolution while keeping the number of sampling points unchanged, thus allowing the estimation of harmonic phasors under DI interference.

## 3. Proposed Method

### 3.1. Determination of Spectral Interference

The presence of interference between spectral lines can be determined by the phase and amplitude characteristics of the spectral lines [30]. Under the condition that the spectral lines are accurate, the phase difference between adjacent spectral lines is 180°. If the phase difference is not 180°, it means that there is a spectral interference. In practical application, two spectral lines A and B near the spectral peak can be selected, and the phase difference Δ*θ* is calculated.
(4)$$\Delta \theta =\left|\left|\theta_{A}-\theta_{B}\right|-180{}^{\circ}\right|<\varepsilon_{1}$$
where *ε*_1_ is the set error threshold. If (4) is satisfied, it is necessary to further determine whether there is a spectral interference by the amplitude characteristic. If it is not satisfied, then there is a spectral interference. For the determination of the amplitude characteristic, three adjacent spectral lines, A, B and C, near the spectral peak are selected. The corrections are done by A and B as well as B and C, respectively. If the results are the same, there is no spectral interference. Firstly, the ratio of adjacent spectral lines is calculated.
(5)$$v_{1}=y_{A}/y_{B},\;v_{2}=y_{B}/y_{C}$$
where *y*_A_, *y*_B_ and *y*_C_ are the corresponding DFT amplitudes of spectral lines A, B and C, respectively. In this paper, the rectangular window is used as an example. The frequency correction Δ*k* of the rectangular window can be expressed as Δ*k* = (*v* − 1)/(*v* + 1), where Δ*k* is the frequency correction of the spectral line and *v* is the ratio of the amplitudes of two adjacent spectral lines. The frequency corrections Δ*f*_A_ and Δ*f*_B_ of the spectral line A and B can be obtained by bringing *v*_1_ and *v*_2_ into the expression of Δ*k*, respectively. The difference of the two corrections is used as a criterion. Since the difference between the numbers of the spectral lines A and B is 1, the correction difference Δ*P* is calculated.
(6)$$\Delta P=\left|\Delta f_{B}-(\Delta f_{A}+1)\right|<\varepsilon_{2}$$
where *ε*_2_ is the set error threshold. If *ε*_2_ is 0.01, it means that the error of Δ*P* is 0.01 times of the frequency resolution. If (6) is satisfied, there is no spectral interference.

### 3.2. Spectrum Refinement Algorithm

For the spectrum containing DI interference, this paper proposes a spectrum refinement algorithm to improve the frequency resolution while keeping the total sampling time unchanged. Therefore, the spectral peaks of DIs can be distinguished and their parameters can be estimated. The basic idea of the proposed method is to build an autoregressive (AR) model based on the existing sampling sequence. Data extrapolation is performed based on the established AR model, thus the sequence space is expanded. Afterwards, DFT analysis is performed on the expanded sequence space. Therefore, the frequency resolution is improved because the sampling frequency remains unchanged. When the frequency resolution is less than the frequency difference of adjacent interharmonics, the interharmonic parameters can be accurately estimated. The harmonic phasors can then be further estimated.

The AR model for the sequence *x*(*n*) can be expressed by the following equation [31]:(7)$$x(n)=\sum_{i=1}^{p}a(i)x(n-i)+\varepsilon_{n}$$
where *p* is the order of the AR model. *ε*_n_ is a random error term with a mean of 0 and a standard deviation of *σ*. The premise of the AR model is that the sequence must have an autocorrelation property. Considering that harmonic data in power systems have certain autocorrelations [32], the signal can be modeled using (7). First, the order *p* is calculated using the singular value decomposition method. The autocorrelation function *r_x_*(*k*) of the sequence *x*(*n*) is calculated as follows:(8)$$r_{x}(k)=\frac{1}{N}\sum_{n=0}^{N-1}x(n)x(n-k)\;k=1,2,\ldots ,2N-1$$
where *N* is the number of the sampled sequence *x*(*n*). The autocorrelation coefficient of *x*(*n*) can be expressed as *r_x_*(1). The autocorrelation matrix **R** of *x*(*n*) is further calculated.
(9)$$R=\left[\begin{array}{cccc} r_{x}(N) & r_{x}(N-1) & \cdots & r_{x}(1)\\ r_{x}(N+1) & r_{x}(N) & \cdots & r_{x}(2)\\ \vdots & \vdots & \ddots & \vdots \\ r_{x}(2N-1) & r_{x}(2N-2) & \cdots & r_{x}(N) \end{array}\right]$$

The matrix **R** is diagonalized using the singular value decomposition as follows:(10)$$R=U\left[\begin{array}{cccc} \lambda_{1} & & & \\ & \lambda_{2} & & \\ & & \ddots & \\ & & & \lambda_{N} \end{array}\right]V^{*}$$
where **U** and **V** are unitary matrices. *λ*_1_, *λ*_2_,…, *λ_N_* are the eigenvalues of matrix **R** and satisfy *λ*_1_ ≥ *λ*_2_ ≥ … ≥ *λ_N_* ≥ 0. Then, the order of the AR model can be obtained by the following equation:(11)$$\left\{ \begin{array}{l} p=\min(m)\\ s.t.{\left[\frac{\lambda_{1}{}^{2}+\lambda_{2}{}^{2}+\ldots +\lambda_{m}{}^{2}}{\lambda_{1}{}^{2}+\lambda_{2}{}^{2}+\ldots +\lambda_{N}{}^{2}}\right]}^{\frac{1}{2}}\geq T_{h}\;,\;m=1,2,\ldots ,N \end{array}\right.$$
where *T*_h_ is the set threshold value. Next, the AR model coefficients *a*(*i*) and the random error *σ*^2^ are calculated. The forward prediction error *f ^p^*(*n*), and the backward prediction error *g^p^*(*n*) are defined as follows:(12)$$\left\{ \begin{array}{l} f^{p}(n)=x(n)+\sum_{i=1}^{p}a(i)x(n-i)\\ g^{p}(n)=x(n-p)+\sum_{i=1}^{p}a(i)x(n-p+i) \end{array}\right.$$

The energy of the error should be minimized when the forward and backward prediction is optimal. In addition, it is necessary to satisfy the orthogonality of the forward and backward prediction errors with the signal. Based on this, the objective functions *P*_1_(*k*), *P*_2_(*k*) and *P*_3_(*k*) are defined as follows:(13)$$\left\{ \begin{array}{l} P_{1}(k)={\left|f^{k}(n)\right|}^{2}+{\left|g^{k}(n)\right|}^{2}\\ P_{2}(k)=f^{k}(n)x(n-i)\;\;\;\;\;\;\;\;\;\;\;1\leq i\leq p\\ P_{3}(k)=g^{k}(n)x(n-i+1)\;\;\;\;1\leq i\leq p \end{array}\right.$$
where *k* is an iterative variable, *k* = 1, 2,…, *p*. The expressions for *f*^*k*^(*n*) and *g^k^*(*n*) are obtained by replacing *p* with *k* in (12). The objective function *J*(*k*) is constructed as follows:(14)$$\begin{array}{l} J(k)=\frac{1}{N-k}\sum_{n=k}^{N-1}P_{1}(k)+\sum_{i=1}^{k}{\left[\frac{1}{N-k}\sum_{n=k}^{N-1}P_{2}(k)\right]}^{2}\\ \;\;+\sum_{i=1}^{k}{\left[\frac{1}{N-k}\sum_{n=k}^{N-1}P_{3}(k)\right]}^{2} \end{array}$$

The estimates of *a*(*i*) and *σ*^2^ are obtained by the minimization of *J*(*k*). According to the Levinson recursive algorithm, the reflection coefficients *β*_1_, *β*_2_,…, *β_p_* are introduced. The recursive formulas of *a*(*i*) and *σ*^2^ can be expressed as follows:(15)$$\left\{ \begin{array}{l} a_{k}(i)=\left\{ \begin{array}{l} a_{k-1}(i)+\beta_{k}a_{k-1}^{*}(k-i)\;\;i=1,2,\ldots ,k-1\\ \beta_{k}\;\;\;\;\;\;\;\;\;\;\;\;\;\;\;\;\;\;\;\;\;\;\;\;\;\;\;\;\;\;\;\;\;\;\;\;\;\;i=k \end{array}\right.\\ \sigma_{k}^{2}=(1-{\left|a_{k}(k)\right|}^{2})\sigma_{k-1}^{2} \end{array}\right.$$
where *a_k_*(*i*) represents the value of the *k*-th iteration of *a*(*i*). To obtain the estimates of the reflection coefficients, the following equation can be derived using the recurrence of the forward and backward prediction errors.
(16)$$\left\{ \begin{array}{l} f^{k}(n)=f^{k-1}(n)+\beta_{k}g^{k-1}(n-1)\\ g^{k}(n)=g^{k-1}(n-1)+\beta_{k}f^{k-1}(n) \end{array}\right.$$

Substituting (16) into (14), and let the partial derivative of *J*(*k*) with respect to *β_k_* to be 0. Thus, the estimate of *β_k_* is obtained. The initial values for the iterations in (14) are
(17)$$\left\{ \begin{array}{l} f^{0}(n)=g^{0}(n)=x(n)\\ \sigma_{0}^{2}=\frac{1}{N}\sum_{n=1}^{N}{\left|x(n)\right|}^{2} \end{array}\right.$$

In practical application, let *k* = 1, after which *β_k_*, *a_k_*(*i*), *σ_k_*^2^, *f^k^*(*n*) and *g^k^*(*n*) are computed in turn. Then, let *k* = *k* + 1, and repeat the above steps until *a_p_*(*i*) and *σ_p_*^2^ are calculated, which are the AR model coefficients *a*(*i*) and the random error *σ*^2^. After obtaining the AR model for the sequence *x*(*n*), data extrapolation can be performed. The additional number of sequence elements to be predicted, Δ*N*, can be expressed as
(18)$$\Delta N=\frac{f_{s}}{\Delta f^{\prime }}-\frac{f_{s}}{\Delta f}=\frac{f_{s}(\Delta f-\Delta f^{\prime })}{\Delta f\Delta f^{\prime }}$$
where *f*_s_ is the sampling frequency. Δ*f* and Δ*f*′ denote the frequency resolution before and after refinement, respectively. The expanded sequence is DFT analyzed and the interharmonic parameters are calculated, thus eliminating the interference of DIs on the harmonic phasor estimation.

### 3.3. Harmonic Phasor Estimation

After eliminating the interharmonic interference, the signal *s*(*t*) is sampled at a sampling interval *T*_s_ = 1/*f*_s_. Then, the discrete representation of the signal *s*(*t*) is [33]
(19)$$s(n)=\sum_{h=1}^{H}\sum_{k=0}^{K_{h}}\frac{{\left(nT_{s}\right)}^{k}}{k!}\left[\frac{p_{h}^{\left(k\right)}}{\sqrt{2}}e^{j\omega_{h}nT_{s}}+\frac{p_{h}^{*(k)}}{\sqrt{2}}e^{-j\omega_{h}nT_{s}}\right]$$
where *ω_h_* = 2π*f_h_* is the actual angular frequency in radian units. Equation (19) can be further written in a matrix form:(20)$$S=\frac{1}{2}\left[\begin{array}{cc} \Phi_{k,h} & \Phi_{k,h}^{*} \end{array}\right]\left[\begin{array}{l} P_{k,h}\\ P_{k,h}^{*} \end{array}\right]=\frac{1}{2}\Phi P$$
where **S** is the *N* × 1 column vector containing *N* input signals. **Φ***_k,h_* = [*φ*_0,1_ *φ*_1,1_ … *φ_Kh_*_,1_ *φ*_0,2_ *φ*_1,2_ … *φ_Kh_*_,2_ … *φ*_0,*H*_ *φ*_1,*H*_ … *φ_Kh_*_,*H*_] is a *N* × *H*(*K_h_* + 1) order matrix. $\varphi_{k,h}=({\left(nT_{s}\right)}^{k}/\sqrt{2}k!)e^{j\omega_{h}nT_{s}}$ is the *N* × 1 order column vector. **P***_k_*_,*h*_ = [*p*_1_^(0)^ *p*_1_^(1)^ … *p*_1_^(*Kh*)^ *p*_2_^(0)^ *p*_2_^(1)^ … *p*_2_^(*Kh*)^ … *p_H_*^(0)^ *p_H_*^(1)^ … *p_H_*^(*Kh*)^] is the *H*(*K*_h_ + 1) × 1 order column vector of TF coefficients. The estimation of the matrix **P** can be obtained by the least squares method:(21)$$\hat{P}=2{\left(\Phi^{H}\Phi \right)}^{-1}\Phi^{H}S$$
where *^H^* is the Hermitian operator. After obtaining the estimate of **P**, the estimate of the harmonic phasor *P_h_*(*t*) within the observation window can be obtained by Taylor formula. Based on the obtained TF coefficients, the derivative estimates of *P_h_*(*t*) can also be obtained as follows:(22)$${\hat{P}}_{h}^{\left(m\right)}(t)|{}_{t=\frac{T_{w}}{2}}=\sum_{k=0}^{K_{h}}\frac{d^{m}}{dt^{m}}\frac{t^{k}}{k!}{\hat{p}}_{h}^{\left(k\right)}|{}_{t=\frac{T_{w}}{2}}$$
where ${\hat{P}}_{h}^{\left(m\right)}(t)$ is the estimated value of the *m*-th order derivative of *P_h_*(*t*), ${\hat{p}}_{h}^{\left(k\right)}$ is the estimated value of the TF coefficient. *T*_w_ is the length of the observation window. In this paper, the estimates of *f_h_* and ROCOF*_h_* are calculated from the estimates at the midpoint of the observation window.
(23)f^h(Tw2)=hf0+12πIm[d1h×d0h*]|d0h|2
(24)ROC^OFh(Tw2)=12πIm[d2h×d0h*]|d0h|2−1πRe[d1h×d0h*]Im[d1h×d0h*]|d0h|4
where *d*_0*h*_, *d*_1*h*_ and *d*_2*h*_ are the zero-order, first-order and second-order derivative estimates of *P_h_*(*t*), respectively. Re and Im are the operations to extract the real and imaginary parts of the complex numbers, respectively. In summary, the proposed harmonic phasor estimation method considering the DI interference is shown in Figure 2.

## 4. Simulation Analysis

In this section, the performance of harmonic phasor estimation of the proposed method and other methods are compared and analyzed under different simulation conditions. The comparison methods are as follows: Method 1 is IpDFT, Method 2 is the Prony algorithm, Method 3 is CS-TFM, and Method 4 is SIFE. The proposed method is considered as Method 5. According to the evaluation criteria for PMU in IEEE Std C37.118.1-2011, this paper uses the total vector error (TVE), frequency error (FE), and ROCOF error (RFE) to measure the effectiveness of harmonic phasor estimation. To simulate the worst scenarios, this paper refers to the measurement standards of both P and M class PMUs. The simulation scenarios include: steady-state test, dense interharmonics test, wideband noise test, frequency deviation test, dynamic-state test and experimental signal test.

### 4.1. Steady-State Test

The test signal can be expressed by the following equation:(25)$$s(t)=\cos(2\pi f_{0}t+\varphi_{1})+\sum_{h=2}^{13}0.1\cos(2\pi hf_{0}t+\varphi_{h})$$
where *f*_0_ is set to 50 Hz, *φ*_1_ and *φ_h_* are set to random numbers uniformly distributed between (0, 2π). The amplitude of each harmonic is set to 10% of the fundamental amplitude. According to [23,34], harmonics after the 13th order in the utility grid have a low proportion of the fundamental. Therefore, the maximum number of harmonics in the simulation is 13. The sampling frequency is 5 kHz and the length of the window function is 6 fundamental periods. In the subsequent simulation analysis, the length of the window function will be kept constant at 0.12 s. Because it is suitable for the dynamic characteristics of varying network signals. The signals shown in (25) are detected using Methods 1 to 5, respectively. Figure 3 shows the maximum TVEs, FEs, and RFEs for each method, where the harmonic order of 1 indicates the fundamental wave.

As shown in Figure 3, all the methods can provide estimates with errors close to zero at nominal frequencies. The maximum TVE and FE limits in [34] for the synchrophasor measurement are 1% and 0.005 Hz, respectively, when harmonic interference is included. The errors of the proposed method satisfy the requirements completely. Since the signal model contains multiple harmonics, Method 1 has problems of the spectrum leakage, fence effect and mutual harmonic interference. Method 2 requires a large number of exponential functions to fit the signal, and it is difficult to design the corresponding filter. Method 3 uses the second-order Taylor formula to model the phasors, so there is a certain amount of model error. Increasing the order of the Taylor model will improve the accuracy, but the higher order increases the computational burden. Method 4 is difficult to determine the coefficients of the sinc interpolation functions at a high harmonic order. Only Method 5 gives the most accurate estimation results.

### 4.2. Dense Interharmonics Test

The test signal can be expressed by the following equation:(26)$$s(t)=\sum_{i=1}^{6}a_{i}\cos(2\pi f_{i\;}t+\varphi_{i})$$

The sampling frequency *f*_s_ is 5 kHz. The number of sampling points is 600. The specific signal parameters are shown in Table 1. From Table 1, the signal contains fundamental, second and third harmonics. Interharmonic interference exists near the fundamental and second harmonic. Figure 4 is the spectrum of *s*(*t*) obtained after the DFT.

In Figure 4, only three spectral peaks can be observed, corresponding to the frequencies of 51.2 Hz, 102.4 Hz, and 153.6 Hz. Not only the spectral peaks of the interharmonics cannot be detected, but also the frequency estimations of the harmonics are biased. The autocorrelation coefficient of the above signal was calculated to be 0.997. Therefore, the AR model can be used for sequence extrapolation. In Figure 4, the frequency resolution Δ*f* = *f*_s_/*N* ≈ 8.33 Hz. In this paper, the refined frequency resolution Δ*f*′ was set to 0.1 Hz. The proposed spectrum refinement algorithm is applied to *s*(*t*), and the refined spectrum is obtained as shown in Figure 5.

From Figure 5, the refined spectrum can completely show the spectral peaks corresponding to each frequency component. In addition, the amplitudes of the spectral peaks are consistent with the real values, which proves the effectiveness of the proposed method. The final results of the estimations of signal parameters for each method are shown in Table 2.

From Table 2, the interharmonic parameters cannot be obtained by Method 1, and there is a large error in the estimation of the harmonic parameters. This is due to the small amplitude of the interharmonic components, as their spectral peaks are covered by the main and side lobes of harmonics. They also interfere with the spectral peaks of the harmonic phasors. The estimations of harmonics are relatively accurate for Methods 2 to 4, but the errors of interharmonics are larger. For Method 2, the presence of DIs increases the complexity of the model. The signal requires more exponential functions for fitting, and the error increases accordingly. This also has a negative impact on Method 3, which increases the computational complexity. The estimation results of Method 3 and Method 4 are closer to the true values, but there are still some deviations. The TF coefficients, filter coefficients and length of the observation window of Method 3 are difficult to determine. Method 4 only considers the first-order and zero-order derivatives of harmonic phasors, which also has certain model errors. Method 5 uses the autocorrelation of the signal for data extrapolation, which improves the frequency resolution and eliminates the spectral interference. Therefore, only Method 5 gives the most accurate estimation results.

### 4.3. Wideband Noise Test

In practice, the sampled signal often contains a certain amount of wideband noise. The simulation signal used in this part is expressed as follows:(27)$$\begin{array}{ll} s(t)= & \cos(2\pi f_{0}t+\varphi_{1})+\sum_{h=2}^{13}0.1\cos(2\pi hf_{0}t+\varphi_{h})\\ & +\sum_{i=1}^{6}0.01\cos(2\pi f_{i}t+\varphi_{i})+\mathrm{noise} \end{array}$$
where noise is Gaussian white noise with a mean value of 0. *f_i_* is the interharmonic frequency and the values are 48.9 Hz, 51.1 Hz, 52.15 Hz, 65.5 Hz, 103 Hz, 213.4 Hz. *φ*_1_, *φ_h_* and *φ_i_* are set as random numbers uniformly distributed between (0, 2π). Other parameters are consistent with Section 4.2. Two noise conditions are considered: 40 dB SNR and 60 dB SNR, respectively. Synchrophasor and harmonic phasor detection are performed for signal (27). The maximum TVEs, FEs and RFEs for each method are shown in Figure 6.

From Figure 6, the errors of Method 2 increase rapidly, which is because Method 2 is extremely sensitive to noise. For wideband noise, it is impossible to use a suitable exponential function model to fit the signal due to its irregularity. Therefore, accurate estimates of the synchrophasor and harmonic phasors cannot be obtained. The errors of Methods 3 and 4 also increase compared to Section 4.1 and Section 4.2. This is because Methods 3 and 4 tend to use an observation window of shorter length in order to obtain a shorter algorithm response time. Therefore, the presence of noise can seriously interfere with the accuracy of the fitting of the Taylor formula and the sinc interpolation formula. Although Method 1 uses windowing and interpolation, there is no procedures to deal with the noise, and thus the error is increased. In contrast, Method 5 is able to accurately identify harmonic phasors under the condition of wideband noise and has an advantage of anti-noise capability.

### 4.4. Frequency Deviation Test

To evaluate the performance of the proposed method under the condition of frequency deviation, the following signal was used for testing:(28)$$\begin{array}{ll} s(t)= & \cos(2\pi ft+\varphi_{1})+\sum_{i=1}^{6}0.01\cos(2\pi f_{i}t+\varphi_{i})\\ & +\sum_{h=2}^{13}0.1\cos(2\pi hft+\varphi_{h}) \end{array}$$

According to the latest IEEE standard [34], *f* was set to vary from 45 Hz to 55 Hz in a step of 1 Hz. Other parameters were consistent with Section 4.3. For Method 1, the spectrum is scattered throughout the frequency domain for signals with varying frequencies, the spectral peaks cannot be accurately identified. Since Method 1 has high errors in the previous tests, it is not discussed in the subsequent analysis. Figure 7 shows the errors in the estimation of harmonic phasors by the proposed method at different fundamental frequencies. Figure 8 shows the estimation errors of the 1st to 7th harmonic phasors for Methods 2 to 5. The maximum TVEs, FEs and RFEs in Figure 8 are the average values of the errors obtained at different fundamental frequencies.

In [34], the TVE and FE limits of M-class PMU for synchrophasor measurement under the condition of frequency deviation are 1% and 0.025 Hz, respectively. From Figure 7, even if this standard is applied to harmonic phasors, Method 5 can still satisfy this standard in the case of small frequency deviations. This is because Method 5 uses the first and second order derivatives to approximate the dynamic phasors. It has a good dynamic performance and still has a high tracking ability under the frequency deviation. Method 2 still does not overcome the drawbacks described in Section 4.1, Section 4.2 and Section 4.3, and the exponential function model cannot be adjusted to follow the dynamic signal. The observation window used in Method 3 may not contain sufficient harmonic information. Moreover, the varying harmonic and interharmonic components can enhance spectral interference, which generates large estimation errors. According to Figure 8, the estimation errors of Method 4 and Method 5 are relatively close. Further simulation conditions need to be established.

### 4.5. Dynamic-State Test

In this section, frequency ramp and harmonic oscillation tests will be simulated. First, the frequency ramp test is performed. Due to disturbances in the power system, the frequencies of the voltage or current signals may vary linearly. To further compare the SIFE (Method 4) and the proposed method (Method 5), the following signal was used for testing:(29)$$\begin{array}{ll} s(t)= & \cos(2\pi ft+\pi R_{1}t^{2})+\sum_{i=1}^{6}0.01\cos(2\pi f_{i}t+\varphi_{i})\\ & +\sum_{h=2}^{13}0.1\cos(2\pi hft+\pi hR_{1}t^{2}) \end{array}$$
where *R*_1_ is the slope of the fundamental frequency and is set to 1 Hz/s. *f* varies linearly from 45 Hz to 55 Hz. The other parameters are identical to those in Section 4.3. The test results are shown in Table 3, where *h* = 1 represents the fundamental wave.

Where the maximum TVEs, FEs and RFEs are the average values of the errors. In [34], the TVE, FE and RFE limits for M-class PMU under the condition of frequency ramp are 1%, 0.01 Hz and 0.2 Hz/s, respectively. From Table 3, the max TVEs of Method 4 and Method 5 are relatively close, but the max FEs and RFEs of Method 4 are much greater than Method 5. This is because Method 4 uses different center frequencies to design the measurement filters, thus the corresponding filters are selected according to the actual frequencies of the signals. The value of the center frequency is related to the fundamental frequency band and is often taken as the midpoint of the frequency band. However, this relationship is valid only when the fundamental frequency varies linearly. Based on (29), *s*(*t*) contains the squared term of time. The fundamental and harmonic frequencies change nonlinearly with time, thus the center frequency is no longer located at the midpoint of the fundamental frequency band. Therefore, the estimation errors of frequency and ROCOF are increased. In contrast, Method 5 does not complete the measurement through filters, but introduces the AR model, which improves the frequency resolution and resistance to DI interference. It achieves the accurate estimation of harmonic phasors in multifrequency signals.

Next, the harmonic oscillation test is performed. The amplitude and phase of harmonics may oscillate due to load variations or subsynchronous oscillations. To further verify the effectiveness of the proposed method, the following signal was used for testing:(30)$$\begin{array}{ll} s(t)= & \left[1+s_{m}(t)][\cos(2\pi f_{0}t+s_{m}(t)+\varphi_{1}\right)\\ & +\sum_{h=2}^{13}0.1\cos(2\pi hf_{0}t+hs_{m}(t)+\varphi_{h})]\\ & +\sum_{i=1}^{6}0.01\cos(2\pi f_{i}t+\varphi_{i}) \end{array}$$
where *s*_m_(*t*) = 0.1cos(2π*f*_m_*t*) and *f*_m_ is the modulation frequency, which is set to 2 Hz. Equation (30) is equivalent to the mathematical expression of the modulated signal. Other parameters are same as in Section 4.3. The test results are shown in Table 4.

As shown in Table 4, the errors of both Methods 4 and 5 increase under the condition of harmonic oscillation, with Method 4 showing a higher error increase. In [34], the TVE, FE and RFE limits are 3%, 0.06 Hz and 2.3 Hz/s, respectively, for P-class PMUs with a reporting rate of 50 Hz. Both Methods 4 and 5 are able to satisfy the TVE standard. For the estimation of synchrophasor and low harmonic phasors, Method 5 is able to satisfy the FE and RFE standard, but Method 4 cannot. In addition to the reasons mentioned above, under the harmonic oscillation condition, Method 4 requires extensive simulations to determine the parameters of each harmonic phasor. Moreover, when the fundamental frequency is 50 Hz, Method 4 cannot obtain results identical to the real value. In summary, Method 5 has a better accuracy and interference immunity compared with other methods.

### 4.6. Experimental Signal Test

To evaluate the practical value of the proposed method, we built a test platform in laboratory conditions as shown in Figure 9. The experimental signal is first generated using a real-time digital simulation system and further processed by a power amplifier and a signal acquisition device. The sampled data are input to the PC to obtain estimates of harmonic phasors, frequencies and ROCOFs.

The steady-state test was first performed. The signal parameters were set as shown in Table 5, where the phases of the frequency components were set as random numbers uniformly distributed between (0, 2π). The accuracy of the estimation results is measured using the following indexes:(31)$$Error=\max\left(\frac{\left|x-\hat{x}\right|}{x}\times 100\%\right)x:f_{i}\;a_{i}\;\varphi_{i}$$
where *x* and $\hat{x}$ are the true and estimated values of the parameters, respectively. To simulate the environment noise, 40 dB SNR and 60 dB SNR of noise are added to the experimental signal. The signal is estimated using Methods 1 to 5. Figure 10 shows the estimation errors for different methods.

From Figure 10, it is clear that Method 1 cannot obtain estimates of the interharmonics and thus cannot be plotted in Figure 10. Methods 2 to 4 can detect all frequency components. However, since Method 2 uses a series of exponential functions to model the signal, the presence of noise makes the modeling difficult. Since the interference from adjacent frequency components, spectrum leakage and fence effect, the spectral peaks of interharmonics cause severe interference to the spectral peaks of harmonics. Although Methods 3 and 4 use a limited number of data points to obtain derivative estimates of the interharmonic phasors, they cannot restore distorted harmonic phasors, so there are still large errors. In contrast, Method 5 eliminates the interharmonic interference by building an AR model. The parameter estimates of Method 5 are almost the same as the real values even under large environment noise.

To analyze the situation under frequency deviation, the fundamental frequency is set to 52.5 Hz and 55 Hz, respectively. The signal difference (*SD*) between the real signal and the reconstructed signal is used as the performance index. *SD* = |*s*(*t*) − *s*_r_(*t*)|, where *s*(*t*) is the experimental signal and *s*_r_(*t*) is the reconstructed signal. Figure 11 shows the waveforms of the experimental signals at different fundamental frequencies and the *SD*s obtained by the proposed method.

As shown in Figure 11, with the increase of the frequency deviation, the estimation errors of the proposed method are within an acceptable range. Therefore, for unknown signals, the proposed method can be applied to estimate harmonic phasor parameters.

## 5. Conclusions

As a result of the time-domain windowing of the signal, the spectrums of signal components with close frequencies overlap each other, causing severe spectral interference and reducing the estimation accuracy of harmonic phasors. Although the frequency resolution can be improved by using a longer observation window, this reduces the ability to track the signal dynamically. To address this problem, this paper proposes a harmonic phasor estimation method that considers DI interference. Firstly, the characteristics of the interfered spectral lines are used to determine whether spectral interference occurs in the signal. After that, data extrapolation is performed based on the signal sampling sequence to refine the spectrum and eliminate the interharmonic interference. Finally, the estimates of harmonic phasors, frequencies and ROCOFs are obtained. The proposed method can remove the interference of DIs while keeping the observation time unchanged. Simulation analysis indicates that the proposed method has a high accuracy under wideband noise, frequency deviation and dynamic conditions (frequency ramp and harmonic oscillation). A potential limitation is that the proposed algorithm does not consider the decaying DC component in the signal. Its presence may affect the estimation accuracy of harmonic phasors. Another limitation is whether the proposed spectrum refinement algorithm has a high computational burden. Therefore, further directions are the estimation of harmonic phasors containing decaying DC components and the computational burden.

## Figures and Tables

**Figure 1 entropy-25-00236-f001:**
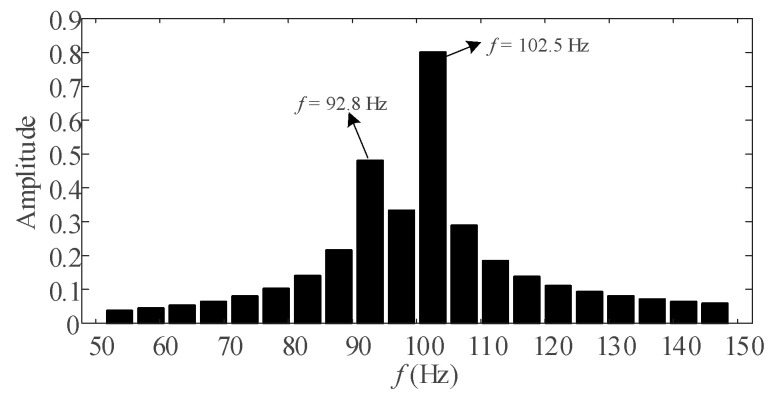
Spectrum of signal *x*(*t*).

**Figure 2 entropy-25-00236-f002:**
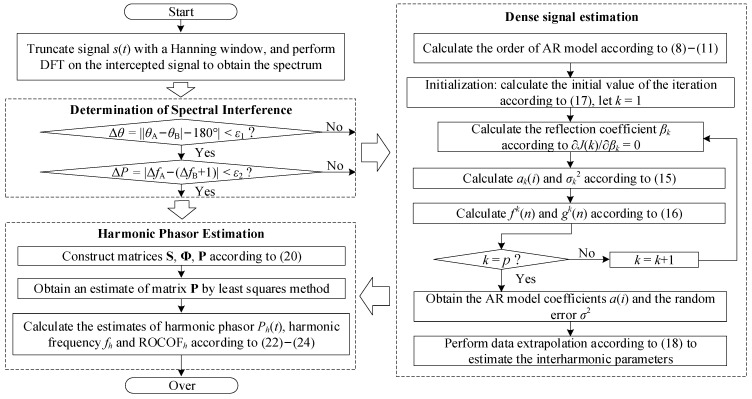
Flow chart of the proposed method.

**Figure 3 entropy-25-00236-f003:**
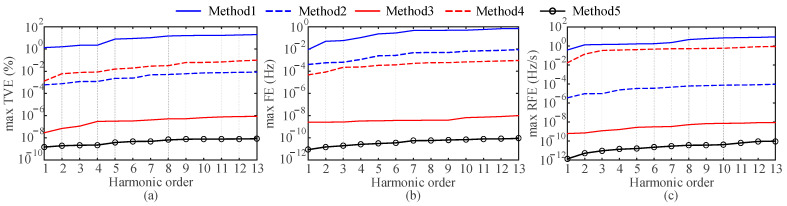
Estimation errors for the 1st to 13th harmonics. (**a**) Max TVE (%), (**b**) max FE (Hz) and (**c**) max RFE (Hz/s).

**Figure 4 entropy-25-00236-f004:**
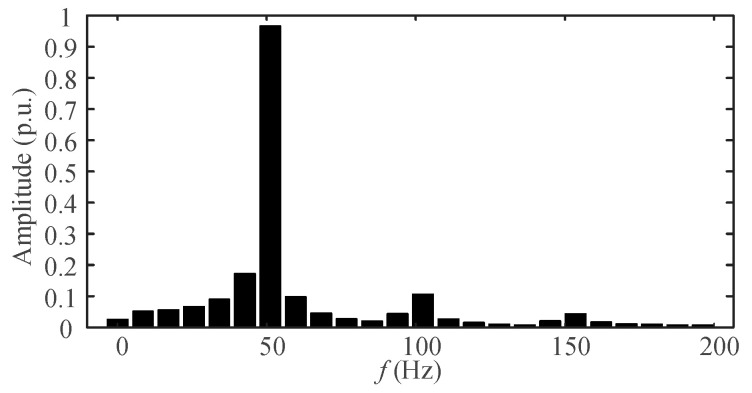
Spectrum of simulation signal.

**Figure 5 entropy-25-00236-f005:**
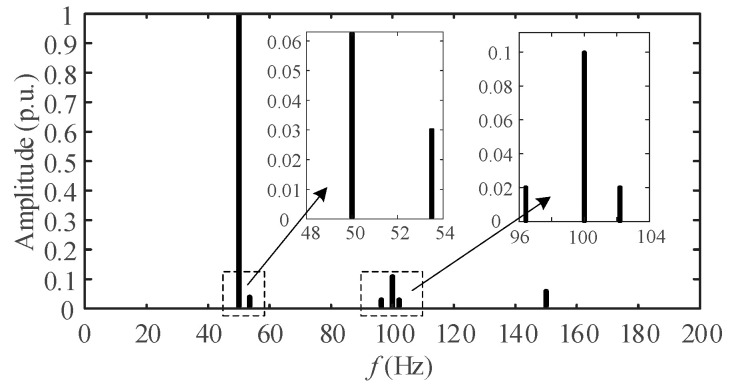
Results of spectrum refinement.

**Figure 6 entropy-25-00236-f006:**
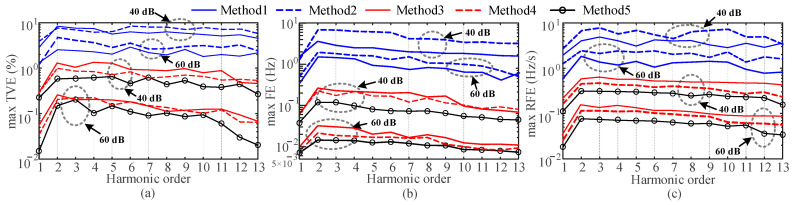
Estimation errors for the 1st to 13th harmonics with SNR = 40 dB and SNR = 60 dB. (**a**) Max TVE (%), (**b**) max FE (Hz) and (**c**) max RFE (Hz/s).

**Figure 7 entropy-25-00236-f007:**
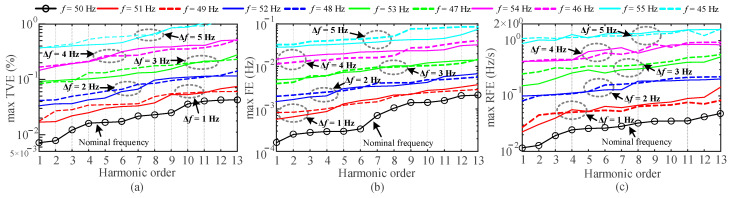
Estimation errors of the 1st to 13th harmonics at different fundamental frequencies (by proposed method). (**a**) Max TVE (%), (**b**) max FE (Hz) and (**c**) max RFE (Hz/s).

**Figure 8 entropy-25-00236-f008:**
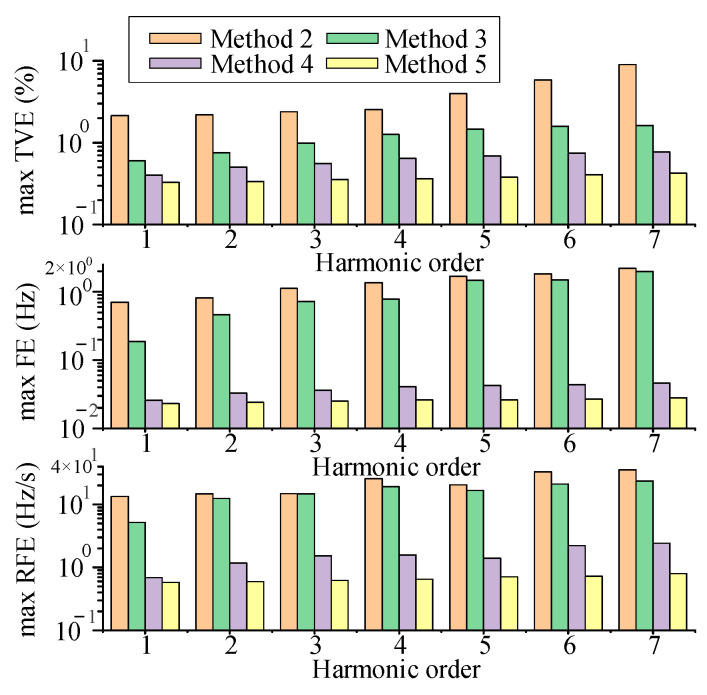
Maximum TVEs, FEs, and RFEs under frequency deviation (by Methods 2 to 5).

**Figure 9 entropy-25-00236-f009:**
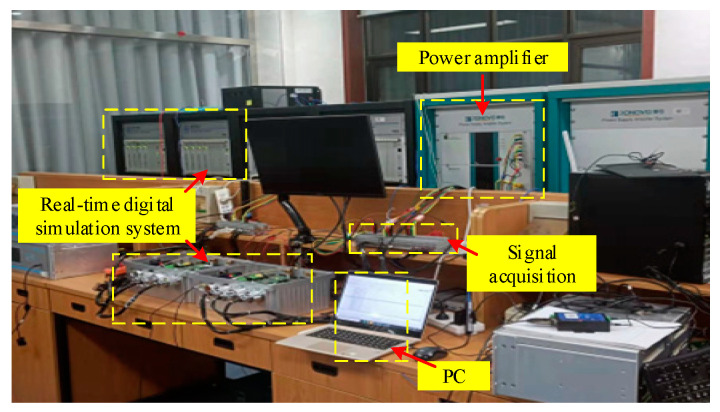
Experimental equipment.

**Figure 10 entropy-25-00236-f010:**
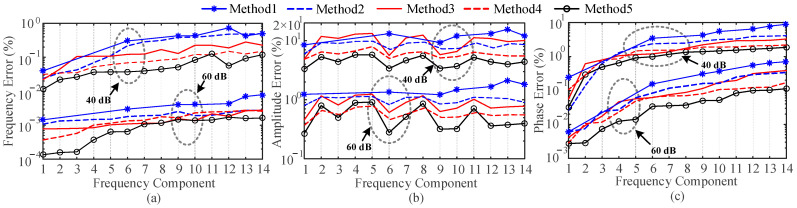
Estimation errors of different frequency components with SNR = 40 dB and SNR = 60 dB. (**a**) Frequency error (%), (**b**) amplitude error and (%) (**c**) phase error (%).

**Figure 11 entropy-25-00236-f011:**
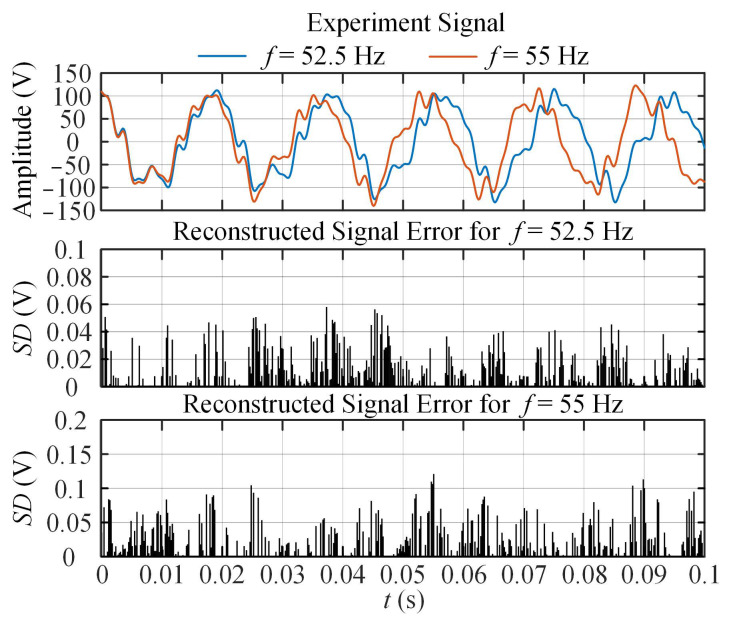
Waveforms of experimental signals and SDs at different fundamental frequencies.

**Table 1 entropy-25-00236-t001:** Simulation signal parameters.

*i*	1	2	3	4	5	6
*f_i_* (Hz)	50	53.5	96.4	100	102.2	150
*a_i_* (p.u.)	1.00	0.03	0.02	0.10	0.02	0.05
*φ_i_* (rad)	0.65	0.37	0.68	0.35	0.87	0.53

**Table 2 entropy-25-00236-t002:** Estimation results of signal parameters.

Method	Actual	1	2	3	4	5
*f_i_* (Hz)	50.00	48.83	50.33	50.02	49.99	50
	53.50	—	55.16	52.71	53.46	53.4
	96.40	—	98.50	97.02	96.42	96.5
	100.00	102.54	99.58	100.15	99.97	100
	102.20	—	103.10	102.68	102.25	102.1
	150.00	151.37	149.45	150.07	149.95	150
*a_i_* (p.u.)	1.00	0.899	0.980	1.001	0.998	1.00
	0.03	—	0.039	0.036	0.029	0.03
	0.02	—	0.011	0.015	0.021	0.02
	0.10	0.082	0.096	0.098	0.101	0.10
	0.02	—	0.025	0.016	0.022	0.02
	0.05	0.055	0.046	0.053	0.048	0.05
*φ_i_* (rad)	0.65	1.40	0.67	0.65	0.65	0.65
	0.37	—	0.33	0.42	0.40	0.39
	0.68	—	0.60	0.58	0.66	0.67
	0.35	−1.14	0.32	0.35	0.35	0.35
	0.87	—	0.69	0.79	0.90	0.88
	0.53	−0.67	0.51	0.54	0.53	0.53

**Table 3 entropy-25-00236-t003:** Maximum TVEs, FEs, and RFEs under frequency ramp condition.

*h*	Max TVE (%)		Max FE (Hz)		Max RFE (Hz/s)	
	Method		Method		Method	
	4	5	4	5	4	5
1	0.178	0.085	0.037	0.005	0.165	0.063
2	0.248	0.031	0.069	0.005	0.206	0.074
3	0.275	0.120	0.114	0.006	0.440	0.089
4	0.386	0.219	0.153	0.006	0.464	0.092
5	0.391	0.364	0.234	0.007	0.534	0.138
6	0.451	0.401	0.277	0.008	0.553	0.186
7	0.643	0.495	0.308	0.009	0.697	0.194
8	0.711	0.501	0.351	0.010	0.705	0.218
9	0.743	0.656	0.391	0.010	0.962	0.230
10	0.810	0.787	0.393	0.015	1.063	0.256
11	0.854	0.792	0.453	0.017	1.283	0.325
12	0.869	0.830	0.455	0.020	1.336	0.329
13	0.945	0.835	0.458	0.022	1.400	0.332

**Table 4 entropy-25-00236-t004:** Maximum TVEs, FEs, and RFEs under harmonic oscillation condition.

*h*	Max TVE (%)		Max FE (Hz)		Max RFE (Hz/s)	
	Method		Method		Method	
	4	5	4	5	4	5
1	0.057	0.048	0.048	0.013	19.578	0.732
2	0.061	0.134	0.133	0.017	18.685	0.955
3	0.357	0.234	0.136	0.020	15.033	1.514
4	0.292	0.271	0.163	0.036	13.749	1.773
5	0.412	0.391	0.180	0.048	12.453	1.820
6	0.917	0.391	0.288	0.054	10.036	1.991
7	1.081	0.569	0.360	0.055	5.778	2.540
8	1.086	0.774	0.363	0.061	2.693	2.769
9	1.106	0.969	0.414	0.061	5.769	2.794
10	1.163	0.913	0.496	0.063	10.003	3.158
11	1.291	1.030	0.523	0.067	13.573	3.818
12	1.301	1.055	0.563	0.077	16.157	3.958
13	1.396	1.108	0.593	0.097	19.233	4.605

**Table 5 entropy-25-00236-t005:** Parameters of the experimental signal.

** *i* **	**1**	**2**	**3**	**4**	**5**	**6**	**7**
*f* (Hz)	50	56	65	83	86	100	102
*A* (V)	100	2.5	3.0	1.0	2.0	10	3.0
** *i* **	**8**	**9**	**10**	**11**	**12**	**13**	**14**
*f* (Hz)	105	150	200	213	250	350	450
*A* (V)	2.5	10	10	3.0	10	10	10

## Data Availability

The data presented in this study are available on request from the corresponding author. The data are not publicly available due to privacy.

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
