# Peer review of "Harmonic Phasor Estimation Method Considering Dense Interharmonic Interference"

_entropy, 2023, doi:10.3390/e25020236_

Round 1

Reviewer 1 Report

Excellent work in which the importance of accurate harmonic phasors measurement in signals such as network signals, in which there may be a high interharmonic content, in addition to the possible variations in frequency, phase and amplitude, is very well highlighted. The proposed method improves on previous methods, which is well illustrated in the article, and does so by improving the frequency resolution without changing the analysis window and thus the time resolution.

The article proposes a very good phasor measurement method, and additionally in the simulations there is a review of other valid but less accurate methods than the proposed method. However, some suggestions for improving the work are as follows:

- Perhaps it would be convenient to add at the end of the abstract that these are not only simulations but also experimental results:

Line 18: “The simulation AND SOME EXPERIMENTAL results demonstrate that the proposed method can accurately estimate the parameters of harmonic phasors when DIs exist in the signal, and has a certain anti-noise capability and dynamic performance”. The case of an experimental signal I don't think it can be considered just another simulation, so to emphasise that it has also worked with real signals it might be good to insist by adding it in the abstract and perhaps even separating case 4.6 in a separate section. Maybe using a real signal generated with a laboratory synthesiser is not as real as analysing a mains signal in a high harmonic content environment, but at least it is a real signal and not just another simulation.

- Please identify, if possible, how the method proposed in this article could be classified, in the final part of the introduction: non-signal-transformation-based versus signal-transformation-based methods, model-based or mode-decomposition-based method, and whether it is static signal estimation or dynamic signal estimation method? or whether it mixes several of these classifications.

- As a general comment, I believe that some citations to bibliographical references supporting the mathematical expressions used in section 3 should be added to facilitate the understanding of the expressions used in the proposed method, except in cases where the expressions are innovations developed by the authors of this article.

- If “In this paper, the Hanning window is used as an example” (line 181), care must be taken with the contribution of the sidebands to each spectral bar, bearing in mind that consecutive spectral bars are being considered. This is corroborated in line 349 of the article when it says "... their spectral peaks are covered by the main and side lobes of the harmonics". Therefore, would it not be better to have used the rectangular window instead of Hanning's window?

- Lines 181-183: “The frequency correction Δk of the Hanning window can be expressed as Δk = (v-2)/(v+1), and the frequency corrections ΔfA and ΔfB of the spectral line A and B can be obtained by bringing v1 and v2 into the expression of Δk respectively”. What are "v", "v1" and "v2", can you explain it better in the text of the article?

- Line 166: “The presence of interference between spectral lines can be determined by the phase and amplitude characteristics of the spectral lines”, but finally the frequency corrections ΔfA and ΔfB are used for the determination of spectral interference (Lines 175-189). So, can you explain why, suddenly, the amplitude criterion becomes a frequency criterion?

- Lines 250-254: “The expanded sequence is DFT analyzed and the interharmonic parameters are calculated, thus eliminating the interference of DIs on the harmonic phasor estimation”. Does it eliminate the interharmonics, or simply by improving the frequency resolution do they have less influence as they are not so close to the harmonics since with the new resolution there are now more spectral bars interleaved between them? Can you please explain very briefly how exactly the interharmonics are eliminated?

- Lin. 279-280: “In summary, the proposed harmonic phasor estimation method considering DI interference is shown as follows...”. It would be clearer to add at the end of the sentence: “... is shown in Figure 2”.

- Lin. 302-303: “length of the window function is 6 fundamental periods”. This corresponds to a window of 6 x 0.02 = 0.12 s, i.e. a resolution of 8.33 Hz. The window of ten cycles for 50 Hz systems (corresponding to approximately 200 ms and a resolution of 5 Hz), as indicated in the IEC 61000-4-7 standard for measurement on network signals, could have been chosen, as this window fits the dynamic characteristics of these network signals. It is also true that in “Application guide to the European Standard EN 50160” it is indicated that the analysis windows can adopt values for “Quasi-stationary harmonics: Tw = 0.1 to 0.5 s”, and for “Fast changing harmonics Tw = 0.08 to 0.16 sec”, so that the revised article's window duration of 0.12 s may be adequate. However, a clarification could be added on lines 302-303 stating that this window has been chosen because it is suitable for the dynamic characteristics of varying network signals with values within those indicated by the standards.

- Then in “4.2. Dense Interharmonics Test” it says that (Line 324) “The sampling frequency fs is 5 kHz. The number of sampling points is 1024.” so now the window length would be 1024/5000 = 0.2048 s which results in a resolution of 4.8828125 Hz as later confirmed in Line 336. Why is the resolution changed now, and why just at that value? Wouldn't it lead to spectral leakage even for harmonic frequencies as this new resolution is not synchronous with them?

- Line 393: Please rewrite the sentence " Since Method 1 has high errors in the previous tests." so that it is connected to the rest of the sentences in the paragraph. As it is written now, it seems to be missing a second consecutive sentence.

- Line 415: Please complete the sentence “Further simulation conditions need to be established.” for a better understanding of why and for what purpose further simulation conditions need to be established.

- At the beginning of section “4.5. Dynamic-State Test” it would be good to make a brief introduction by anticipating that in this section the cases of “frequency ramp” (together with “frequency deviation”) will be simulated, and then at the end of this same section (Lin. 443 onwards) another different case of Dynamic-State Test is simulated but now with “harmonic oscillation”. As it stands now, the simulation between lines 419-442 and the simulation between lines 443-460 seem to be two different sections that have been wrongly mixed up.

- In 4.5 the conditions of section 4.4 “Frequency Deviation” are merged with the new conditions of section 4.5 “Dynamic-State Test”. If so, are the test results shown in Table 3 the average values of the errors obtained at different fundamental frequencies (as in Fig. 8)? If this is correct, please can you clarify that the test results shown in Table 3 are also the average values, for example at Lin. 424-425.

- Please can you briefly clarify what exactly is meant by the term “decaying DC component” in the signal, which is only mentioned in the Conclusion section and not explained elsewhere in the article?

- What about the computational burden of the proposed method-5, compared to the others? I believe that, in addition to the comments in the last section of conclusions on the limitation of the proposed method in terms of “A potential limitation is that the proposed algorithm does not consider the decaying DC component in the signal. Its presence may affect the estimation accuracy of harmonic phasors” (lines 518-519), it would also be useful to indicate, as a second limitation, whether the proposed method involves a higher computational cost than the other methods analysed in this article.

Round 2

Reviewer 1 Report

Your article is now better understood and I hope that this has improved the quality of your publication. Good proofreading work, congratulations.